molecular biology/plant science

*Camellia oleifera*, seed size, phytohormone, gene expression, WGCNA

**Authors for correspondence:**
Wenfang Gong
e-mail: gwf018@126.com
Deyi Yuan
e-mail: yuan-deyi@163.com

†These authors contributed equally to this study.

# Hormone analysis and candidate genes identification associated with seed size in *Camellia oleifera*

Ke Ji[1,†], Qiling Song[1,†], Xinran Yu[1], Chuanbo Tan[2], Linkai Wang[1], Le Chen[1], Xiaofeng Xiang[1], Wenfang Gong[1] and Deyi Yuan[1]

[1]Key Laboratory of Cultivation and Protection for Non-Wood Forest Trees of Ministry of Education and the Key Laboratory of Non-Wood Forest Products of Forestry Ministry, Central South University of Forestry and Technology, Changsha, Hunan, 410004, People's Republic of China
[2]Hunan Great Sanxiang Camellia Oil Co., Ltd, Hengyang, Hunan 421000, People's Republic of China

WG, 0000-0002-9292-7262

*Camellia oleifera* is an important woody oil species in China. Its seed oil has been widely used as a cooking oil. Seed size is a crucial factor influencing the yield of seed oil. In this study, the horizontal diameter, vertical diameter and volume of *C. oleifera* seeds showed a rapid growth tendency from 235 days after pollination (DAP) to 258 DAP but had a slight increase at seed maturity. During seed development, the expression of genes related to cell proliferation and expansion differ greatly. Auxin plays an important role in *C. oleifera* seeds; *YUC4* and *IAA17* were significantly downregulated. Weighted gene co-expression network analysis screened 21 hub transcription factors for *C. oleifera* seed horizontal diameter, vertical diameter and volume. Among them, *SPL4* was significantly decreased and associated with all these three traits, while *ABI4* and *YAB1* were significantly increased and associated with horizontal diameter of *C. oleifera* seeds. Additionally, *KLU* significantly decreased (2040-fold). Collectively, our data advances the knowledge of factors related to seed size and provides a theoretical basis for improving the yield of *C. oleifera* seeds.

# 1. Introduction

*Camellia oleifera* is a small evergreen tree or shrub of the *Camellia* genus, Theaceae [1]. It is known as one of the world's four

largest woody oil plants with oil palm, olive and coconut [2]. The seed of *C. oleifera* has high oil content with nutrition and healthcare value [3–5]. However, the low yield of *C. oleifera* seeds has been seriously restricting the healthy development of the *C. oleifera* industry [6]. Manipulating seed size is an ideal way to solve this problem, because it is reported that seed size is a crucial factor influencing the final seed yield [7]. Currently, some genes have been identified to play a role in determining seed size in *Arabidopsis thaliana* and rice (*Oryza sativa*). However, no such studies have been reported on *C. oleifera*.

The growth and development of seeds begin after pollination and fertilization, and pass through two important periods of cell proliferation and cell expansion [8]. Cell proliferation and cell expansion play key roles in seed growth and development because they determine the number and size of cells in seeds and ultimately affect seed size [9,10]. Among them, the different cell cycles in cell proliferation are regulated by a protein complex composed of cell cycle-dependent kinases (CDKs) and cyclins [11–13]. Current research mainly focused on the regulation of the cell cycle by A- and B-type CDKs (*CDKAs* and *CDKBs*) [14,15] and A-, B- and D-type cyclins (*CYCAs*, *CYCBs* and *CYCDs*) [16–18]. However, cell expansion drives the extension of the cell wall and further affects cell size [19]. Therefore, cell size is closely related to the extension of cell wall. Enzymes acting on the cytoskeleton play key roles in cell wall extension [20]. Among them, expansins (EXPs) and xyloglucan endotransglucosylase/hydrolase (XTHs) are important participants in the regulatory framework of the cell wall [21]. Expansins disrupt hydrogen bonding between hemi cellulosic wall components and cellulose microfibrils to facilitate cell wall extension [22]. Unlike expansin, XTHs catalyse the connection of xyloglucan molecules to itself and specifically hydrolyse xyloglucan $\beta$-1,4 glycosidic bonds [23,24].

In addition to cell proliferation and cell division, the HAIKU (IKU) pathway, the ubiquitin-proteasome pathway, G-protein signalling pathway and mitogen-activated protein kinase (MAPK) signalling pathways have been discovered to control seed size in *Arabidopsis* and rice [25]. The ubiquitin-proteasome pathway, G-protein signalling pathway and MAPK signalling pathways have been shown to control seed size by regulating the growth of maternal tissues, while the IKU pathway controls seed size by regulating zygotic tissues [26,27]. Furthermore, phytohormones related to regulating seed size have also been studied, Brassinolide (BR), auxins and salicylic acid (SA) are considered to be important participants [26,28]. In rice, mutants with defects in BR biosynthesis or signal transduction showed shortened seed length, and the overexpression of the BR biosynthesis gene in rice increased seed size [29,30]. This indicates that BR plays an important role in seed size. Auxin has been studied to regulate rice seed size by setting physical constraints via the seed coat [31]. Auxin response factors 2 *(ARF2)/ MEGAINTEGUMENTA (MNT)* is a repressor of cell division and organ growth, and it controls seed size by limiting the cell proliferation of integument [32]. Compared with wild-type seeds, *mnt/arf2* mutants increased seed and organ size [26]. SA has also been studied to increase rice yield [33] and regulate cell growth in *Arabidopsis* [28]. Seed size is also regulated by transcription factors (TFs) [26], such as *APETALA 2 (AP2)*, *KLUH (KLU)* and *ENHANCER3 OF DA1 (EOD3)*.

Current research on *C. oleifera* seeds mainly focuses on the genes related to oil content and fatty acid composition [34,35], but there are a few studies on seed size genes. In this study, the content of phytohormone was detected. Gene expressions associated with cell proliferation and cell cycle, cell expansion, known signal pathways and phytohormone at different stages were analysed. Weighted gene co-expression network analysis (WGCNA) was used to analyse the correlation between seed size of *C. oleifera* and TFs. This mainly provides a theoretical basis for improving the yield of *C. oleifera* seeds.

# 2. Material and methods

## 2.1. Plant materials collection

The plant materials used in this study came from 6-year-old, high-crown-grafted *C. oleifera* cultivars 'Huashuo' that had reached their peak production stage. The sampling site was in the Dongcheng Camellia experimental base in Wangcheng District, Changsha City, Hunan Province (28°05′ N, 113°21′ E), which has a humid mid-subtropical monsoon climate, the annual average temperature is 17°C and the precipitation is 1370 mm. Fifteen trees without significant pests or diseases, that were growing well and had undergone consistent management measures, were selected randomly. The collection of samples was randomly selected from four different directions of 15 trees and repeated three times. The fruits were collected at 210 days after pollination (DAP) (T1), 235 DAP (T2), 258 DAP (T3), 292 DAP (T4) and 333 DAP (T5). The seeds were frozen and stored at −80°C for analysis.

## 2.2. The determination of seed size and volume

Ten *Camellia* seeds were randomly selected and their vertical diameter and horizontal diameter were measured using a 1/100 vernier caliper. Seed volume was measured according to the method of Li *et al*. [36]. Briefly, 50 ml water was added to the 100 ml measuring cylinder, and the recording volume was V1. After adding five seeds, the recording volume was V2, so the volume of each seed was V2 - V1/5. When the seeds were small (235 DAP and 258DAP), the seed volume was measured with a 10 ml measuring cylinder.

## 2.3. Analysis of phytohormone content

The determination of phytohormone content refers to the method of Song *et al*. [37]. Grind 0.5 g of frozen sample was ground with liquid nitrogen and placed in a centrifuge tube. Phosphate buffered saline buffer (pH = 7.4) of nine times the sample volume was added, then centrifuged for 30 min (4°C, 8000 r.p.m.). The supernatant was stored at 4°C for later use. This was repeated three times. The contents of indole-3-acetic acid (IAA), BR and SA were determined using an enzyme-linked immunosorbent assay kit according to manufacturer's instructions.

## 2.4. Co-expression network construction and hub genes selection

All genes were obtained from published Illumina RNA-Seq data of *C. oleifera* seeds in the National Center for Biotechnology Information database (SRA with accession PRJNA668531; https://dataview.ncbi.nlm.nih.gov/object/PRJNA668531). The genes were screened based on the non-redundant protein database. Weighted gene correlation network analysis was used to construct a co-expression network according to Gong *et al*. [35] through the default network construction and module detection method. The power of $\beta$ was set at 8 and the minimum number of module genes was set at 20. The hub gene selection was based on module membership (MM) and intramodular connectivity (K.in) values in the WGCNA results, and the top three genes of MM and K.in value in high correlation modules were selected as hub genes.

## 2.5. Statistical analysis

All graphs were obtained using ORIGIN PRO (version 2018, OriginLab, Northampton, MA), RSTUDIO (v. 1.3.1056, Boston, Massachusetts, USA) and TBTOOLS (v. 1.0983). The variance analysis was performed using SPSS (v. 19.0, Chicago, USA). All data had three biological replicates. To analyse the difference in multiple comparisons, the values of mean ± s.d. was calculated. Significance was defined at the $p < 0.05$ level.

# 3. Results

## 3.1. Variations in *Camellia oleifera* seeds size at different developmental stages

The horizontal diameter, vertical diameter and volume changes in *C. oleifera* seeds at different developmental stages are shown in figure 1. The increasing trends in horizontal diameter, vertical diameter (figure 1*a*) and volume (figure 1*b*) of seeds were similar, showing rapid growth from 235 DAP (T2) to 258 DAP (T3) with a slight increase at seed maturity (T4–T5). From 235 to 258 DAP, the horizontal diameter, vertical diameter and seed volume increased by 26.67%, 23.66%, and 5.55%, respectively, which was significantly higher than that in other periods. These results showed that 235 to 258 DAP was the period of rapid growth of *C. oleifera* seeds (electronic supplementary material, table S6).

## 3.2. Changes in the expression of genes involved in cell proliferation and cell cycle

Genes related to cell proliferation and cell cycle play important roles in the growth and development of plants [10]. As shown in figure 2*a* and the electronic supplementary material, table S1, CDKs and cyclin-related genes involved in cell proliferation and cell cycle of *C. oleifera* seeds were analysed. Among CDKs genes, *CDKB2-2* was highly expressed from T1 to T3, and its expression level in the T5 period was reduced by 93% compared to the T1 period. The cyclin genes in *C. oleifera* seeds were mainly A-, B- and D-type cyclins (*CYCA, CYCB* and *CYCD*). The expression level of most cyclin genes in the T1–T3 period was significantly higher than that in the T4–T5 period. Among cyclin genes, *CYCD3-1*, however, showed

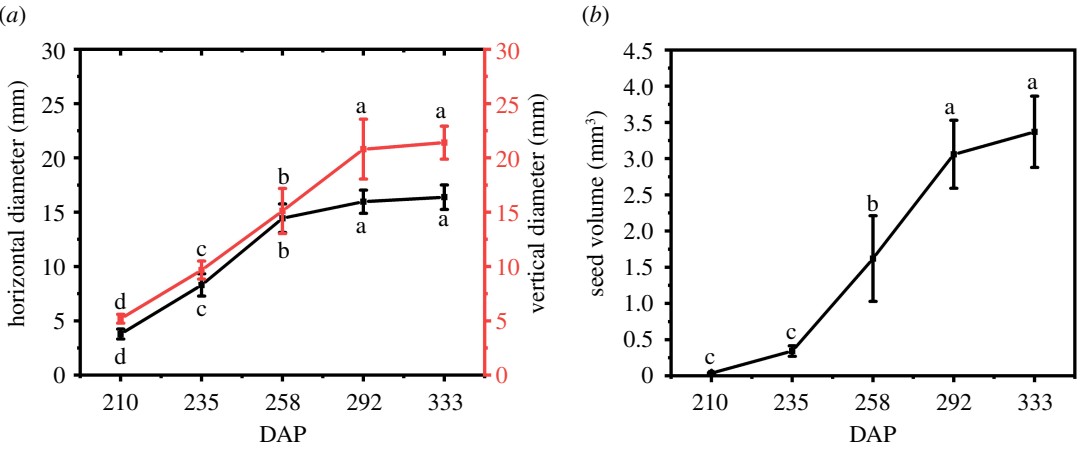

**Figure 1.** Horizontal diameter, vertical diameter and volume of *C. oleifera* seeds in different developmental periods. The dynamic changes of horizontal diameter, vertical diameter (*a*) and seed volume (*b*). The data are the mean ± s.e. of 10 biological replicates and mean values (*n* = 10) followed by different lowercase letters in each column indicated significant differences at *p* < 0.05 by Duncan's multiple range test.

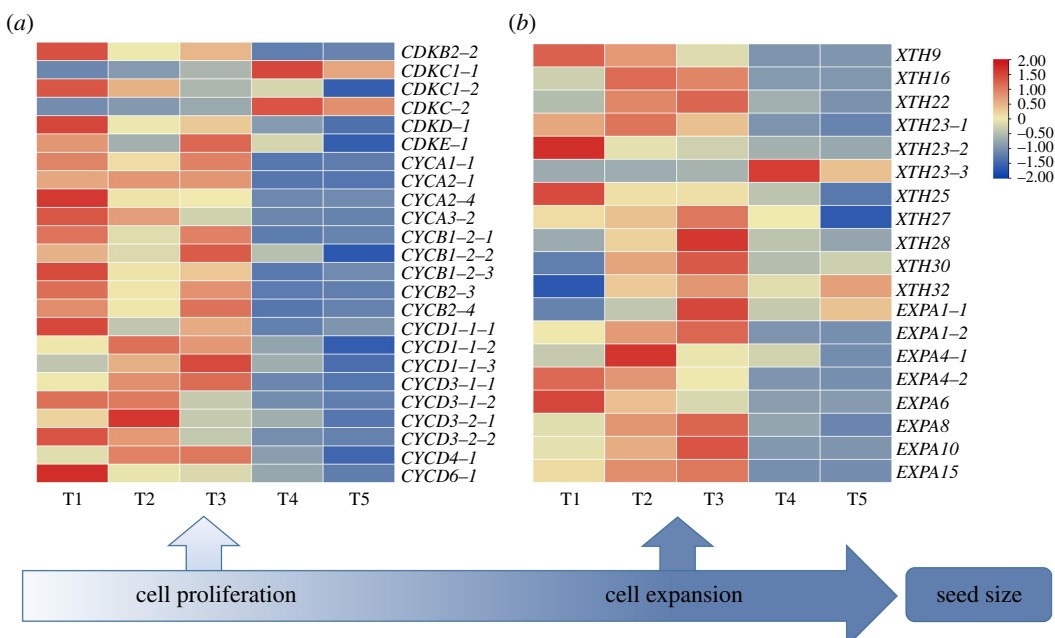

**Figure 2.** Heatmap comparison of genes related to cell proliferation and cell expansion of *C. oleifera* seed. The heatmap shows the expression levels of genes related to cell proliferation (*a*) and cell expansion (*b*) of *C. oleifera* seeds in different periods. The red and blue colours represent upregulated and downregulated, respectively. Abbreviations: *CDKB, CDKC, CDKD, CDKE*: B-, C-, D, E-type cycle-dependent kinases; *CYCA, CYCB, CYCD*: A-, B- and D-type cyclins; *XTHs*, xyloglucan endotransglucosylase/hydrolase genes; *EXPA*, A-type expansins.

different trends. The expression of *CYCD3-1-1* increased from T1 to T3, which was consistent with the change trend of seed size; however, the expression of *CYCD3-1-2* decreased from T1 to T3, which was opposite to the trend change in seed size. Compared to the T1 period, the expression of *CYCD3-1-2* was decreased to 6% in the T5 period. It was speculated that *CYCD3-1* plays different roles in *C. oleifera* seed size.

## 3.3. Expression level analysis of genes involved in cell expansion

The extension of the cell wall affects seed size by regulating cell size [8,19]. As important genes in the cell wall, *EXPs* and *XTHs* gene families were analysed. As shown in figure 2*b* and the electronic

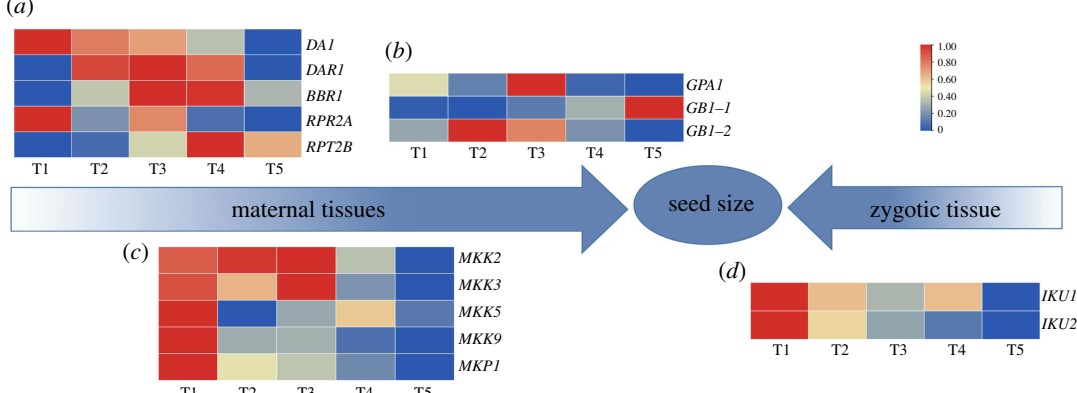

**Figure 3.** The expression of genes involved in signalling pathways related to seed size in *C. oleifera* seed development. The heatmap shows the expression levels of genes related to the ubiquitin-proteasome pathway (*a*), G-protein signalling pathway (*b*), MAPK pathway (*c*), and IKU pathway (*d*) in *C. oleifera* seeds at different stages. Red indicates upregulated genes, and blue indicates downregulated genes. Abbreviations: *DAR1*, DA1-related protein; *BBR*, BIG BROTHER-related; *GPA1*, Gαsubunit 1; *GB1*, Gβsubunit 1; *MKK*, mitogen-activated protein kinase; *MKP*, mitogen-activated protein kinase phosphatase; *IKU*, HAIKU.

supplementary material, table S1, subfamily *EXPA* was the main member of *EXPs* gene family in *C. oleifera* seeds. The expression levels of *EXPA1-2*, *EXPA10* and *EXPA15* were upregulated at first but then reduced at seed maturity (T4–T5). Compared to the T3 period, *EXPA1-2*, *EXPA10* and *EXPA15* were reduced by 99.2%, 99.4% and 99.7% in the T5 period, respectively. The expression level of *EXPA4-2* was reduced by 99.6% in the T5 period compared to the T1 period. Among *XTHs* gene family, *XTH9* was highly expressed in the T1 period, but the expression in the T4 period decreased to 2%. *XTH16* was highly expressed in the T2 period but then decreased. *XTH23* showed different trends, *XTH23-2* decreased from the T1 to T5 period while the expression of *XTH23-3* increased 420-fold in the T4 period compared with the T1 period.

## 3.4. Variations in gene expression in signal pathways associated with the control of seed size

Gene expression levels of the ubiquitin-proteasome pathway, G-protein signalling pathway, MAPK pathway and IKU pathway were analysed in *C. oleifera* seeds. Among the screened genes related to the ubiquitin-proteasome pathway (figure 3*a*; electronic supplementary material, table S2), the expression levels of *DA1* and *PRT2A* showed a downward trend. The expression levels of *DA1* in the T5 period was reduced by 86% compared to the T1 period. However, the expression levels of *DAR1*, *BBR* and *PRT2B* increased first and then decreased. In the G-protein signalling pathway (figure 3*b*; electronic supplementary material, table S2), the expression level of *GB1-1* in the T5 period was upregulated by fourfold compared to the T1 period, while *GB1-2* was highly expressed in the T2 period. Additionally, genes related to the MAPK pathway were analysed (figure 3*c*; electronic supplementary material, table S2). The expression levels of *MKK2*, *MKK3*, *MKK9* and *MKP1* showed a downward trend. The expression levels of *MKP1* were downregulated by 2.7-fold in the T5 period compared to the T1 period. Among the screened genes related to the IKU pathway (figure 3*d*; electronic supplementary material, table S2), the expression levels of *IKU1* and *IKU2* in the T5 period was reduced by 50% and 82% compared with the T1 period, respectively.

## 3.5. Changes and correlation analysis of phytohormones content and related gene expression

The content of IAA, SA and BR in *C. oleifera* seeds was determined (figure 4*a*(i), *b*(i), *c*(i)). From the T1 to T3 period, the content of IAA, SA and BR showed an increasing trend, but then significantly decreased from the T4 to T5 period. Genes related to phytohormones biosynthesis and signalling pathways were analysed (figure 4*a*(ii), *b*(ii), *c*(ii); electronic supplementary material, table S3). The expression levels of most auxin-related genes were relatively higher in T1–T3 than in T4–T5 (figure 4*a*(iii)). In the process of auxin biosynthesis, the expression level of *YUC4* in the *YUC* gene family was 1724-fold in the T1 period compared to the T5 period. In the process of the auxin signalling pathway, *IAA17* and *IAA27-1* were highly expressed in the T1 period, but were reduced by 98% and 96% in the T5 period, respectively.

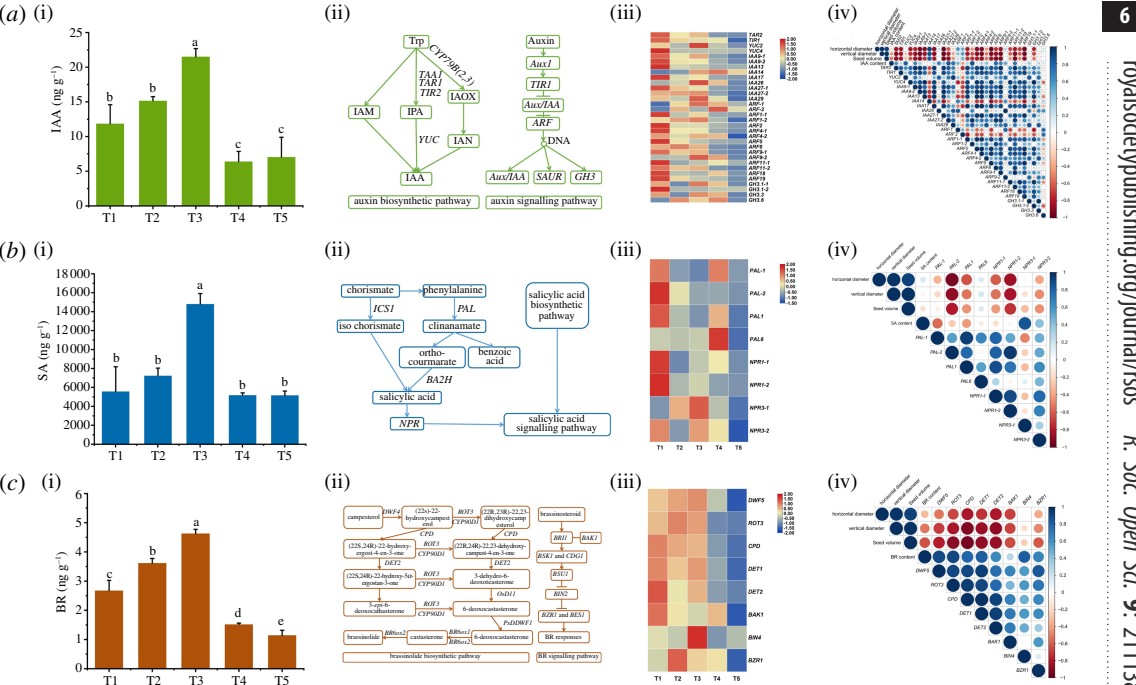

**Figure 4.** Changes and correlation analysis of phytohormone contents and related gene expressions in developmental *C. oleifera* seeds. The content (i) and expression levels of genes (iii) involved in the biosynthetic and signal pathways (ii) of IAA (*a*), SA (*b*) and BR (*c*). Correlation analysis of phytohormone content and related gene expressions with horizontal diameter, vertical diameter and seed volume of *C. oleifera* seeds (iv). The red and blue colours represent upregulated and downregulated, respectively. Abbreviations: *TAR*, tryptophan aminotransferase-related; *TIR*, transport inhibitor response; *YUC*, YUCCA; *AUX/IAA*, auxin/IAA-inducible protein; *ARF*, auxin response factor; *GH*, Gretchen Hagen; *PAL*, phenylalanine ammonia lyase; *NPR*, non-expressor of pathogenesis-related; *DWF*, dwarf; *CPD*, constitutive photomorphogenic dwarf; *BAK*, BRI1-associated receptor kinase; *BIN*, BR-insensitive; *BZR1*, brassinazole resistant 1.

In addition, *IAA14* was upregulated during the development of *C. oleifera* seeds, and in the T4 period, its expression level was 799-fold compared to that in the T1 period. Ten *ARFs* were expressed during the development of *C. oleifera* seeds. Among these *ARFs*, *ARF2* was continuously decreased. Most of the genes in the SA biosynthesis and signalling pathways were expressed at a relatively high level in the T1 period (figure 4*b*(iii)). Compared to the T1 period, *PAL-2*, *PAL1* and *PAL6* related to SA biosynthesis were decreased by 99%, 95% and 95% in the T5 period, respectively. As shown in figure 4*c*(iii), genes related to BR biosynthesis and signalling pathways in *C. oleifera* seeds were expressed at a relatively high level from T1 to T3, but then decreased, which was consistent with the change trend of BR content.

Correlation analysis shows that most auxin-related genes were negatively correlated with seed horizontal diameter, vertical diameter and volume of *C. oleifera* seeds, but positively correlated with IAA content (figure 4*a*(iv)). Among them, *YUC4*, *IAA17*, *IAA27-1*, *ANT* and *ANT17* were significantly negatively correlated with seed horizontal diameter, vertical diameter and volume of *C. oleifera* seeds, but *IAA14* was significantly positively correlated with them. Among the genes related to SA (figure 4*b*(iv)), *PAL-2* and *NPR1-2* were significantly negatively correlated with seed horizontal diameter, vertical diameter and volume of *C. oleifera* seeds. As shown in figure 4*c*(iv), all the screened genes related to BR were negatively correlated with seed horizontal diameter, vertical diameter and volume of *C. oleifera* seeds, but positively correlated with BR content.

## 3.6. Gene expressions of hub genes and selected transcription factors related to seed size

A total of 1880 TFs were identified from the *Camellia* seed transcriptome according to Gong *et al.* [35]. In order to screen for important TFs related to horizontal diameter, vertical diameter and seed volume, WGCNA was performed (figure 5). After removing the TFs with reads per kilobase per million mapped reads (RPKM) = 0 at the detected periods of developing seeds, the remaining 1879 TFs were obtained. The TFs were divided into 13 modules and marked with different colours (figure 5*a*). The number of genes in these modules ranged from 1 to 442 (figure 5*b*). The correlation analysis showed that yellow,

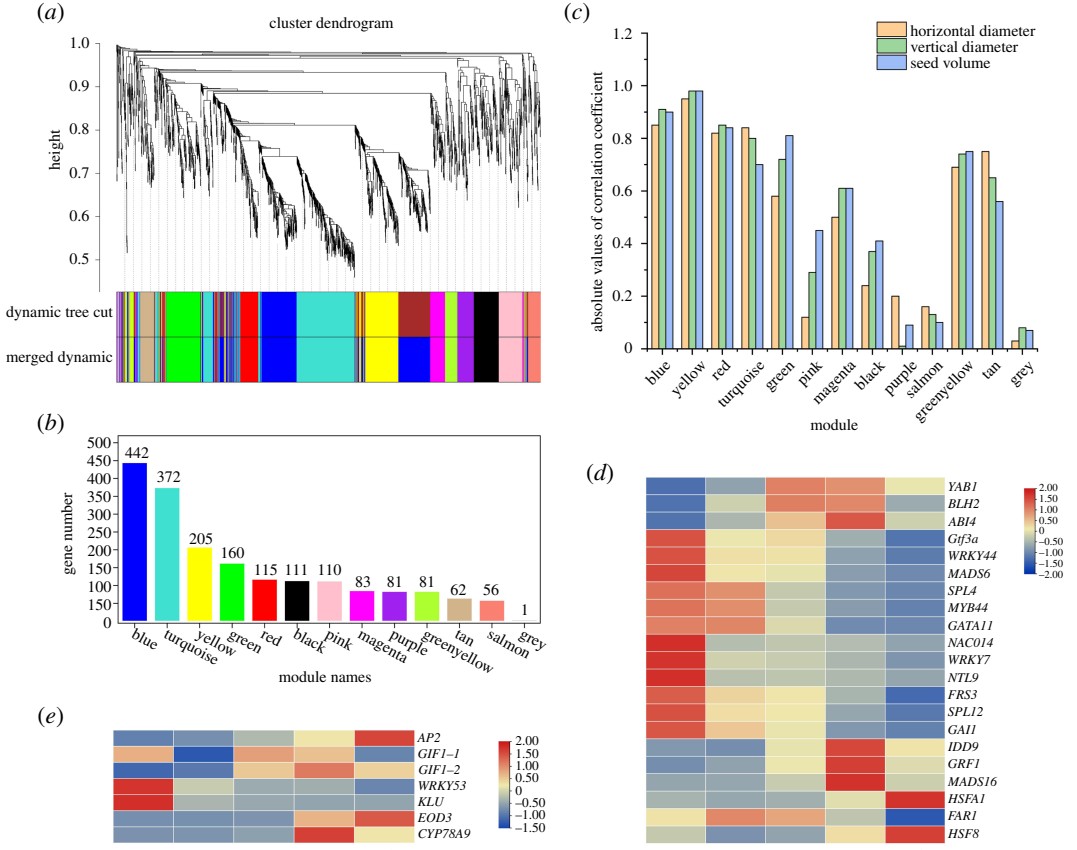

**Figure 5.** WGCNA identifying TFs related to horizontal diameter, vertical diameter and volume of *C. oleifera* seeds. (*a*) Clustering dendrograms of genes. Dissimilarity was based on topological overlap, with assigned module colours. The 13 co-expression modules are shown in different colours. (*b*) Number of genes in 13 modules. (*c*) Correlation between 13 modules and seed horizontal diameter, vertical diameter and volume. (*d*) The expression levels of hub genes in *C. oleifera* seeds at different developmental stages. (*e*) Heatmaps show the expression levels of TFs related to seed size. The red and blue colours represent upregulated and downregulated, respectively. Abbreviations: *YAB1*, YABBY1; *BLH2*, bel1-like homeodomain2; *ABI4*, ABA insensitive 4; *Gtf3a*, transcription factor IIIA; *WRKY*, WRKY transcription factor; *MADS*, MADS-box transcription factor; *SPL*: squamosa promoter-binding-like protein; *MYB*, MYB family protein; *GATA11*: GATA transcription factor; *NAC014*, NAC domain-containing protein 14; *NTL9*, protein NTM1-like 9; *FRS3*, fibroblast growth factor receptor substrate 3; *GAI1*, DELLA protein GAI1; *IDD9*, protein indeterminate-domain 9; *GRF1*, growth-regulating factor 1; *HSF*, heat stress transcription; *FAR1*, protein FAR1-related sequence; *AP2*, APETALA 2; *GIF*, GRF-interacting factor; *KLU*, KLUH; *EOD3*, ENHANCER3 OF DA1.

blue, turquoise, red and tan modules had a higher correlation with horizontal diameter (figure 5*c*; electronic supplementary material, table S4). Yellow, blue, turquoise, red, green and greenyellow had a higher correlation with the vertical diameter. Yellow, blue, red, green and greenyellow had a higher correlation with seed volume (|correlation coefficient| greater than 0.7, *p*-value less than 0.01). The top three genes with high MM and K.in values in each module were selected as hub genes.

As shown in figure 5*d*, 21 hub genes were chosen (electronic supplementary material, table S5). Among them, *Gtf3a*, *WRKY44*, *MADS6*, *SPL4*, *MYB44*, *GATA11*, *FRS3*, *SPL12* and *GAI1* were hub genes common to the horizontal diameter, vertical diameter and seed volume. Compared with the T1 period, all the common hub genes showed the same trend, and the expressions of *SPL4*, *MADS6*, *GAI1*, *MYB44*, *GATA11*, *WRKY44*, *FAR1* and *SPL12* were reduced by 99%, 97%, 96%, 86%, 84%, 79%, 76% and 64% in the T5 period, respectively. Additionally, hub genes *YAB1* and *ABI4* in the tan module related to the horizontal diameter were first significantly upregulated but then decreased. Compared with T1, *YAB1* was increased 800-fold in T3 and *ABI4* was increased 560-fold in T4.

As shown in figure 5*e* and the electronic supplementary material, table S5, the expression levels of known seed size-related TFs were also analysed in *C. oleifera* seeds [26]. The expression levels of *AP2*, *KLU* and *EOD3* performed significantly. Compared to the T1 period, *AP2* and *EOD3* were increased 14-fold and 259-fold in the T5 period, respectively. *KLU* was highly expressed in the T1 period and then decreased. The expression level in T5 was 2040-fold lower than that in the T1 period.

# 4. Discussion

## 4.1. Seed size of Camellia oleifera regulate by cell proliferation, cell expansion and signal pathways

The change in seed size is a complicated process regulated by different genes [13]. As an important process of seed development, cell proliferation and cell expansion affect seed size to a large extent. The regulation of the cell cycle on cell proliferation mainly relies on the complex composed of CDKs and cyclins to control G1/S and G2/M phases [38]. During the development of *C. oleifera* seeds, the expression levels of most *CDKs* and cyclin genes in the T1–T3 period were significantly higher than those in the T4–T5 period. Among them, *CDKB* was unique in plants and only expressed in the G2/M process [39]. In *Arabidopsis* and rice, the degradation of *CDKB2* indirectly influenced the mitotic cycle and led to the presence of polyploid cells [40]. Different from the existence of *CDKB2-1* and *CDKB2-2* in *Arabidopsis* [41], only *CDKB2-2* was highly expressed from the T1 to T3 period of *C. oleifera* seeds. The D-type cyclin *CYCD3-1* restricted the transition of the G1/S phase and overexpressed *CYCD3-1* in leaves leading to excessive proliferation of incompletely differentiated polygonal leaf cells in *Arabidopsis* [42,43]. In *C. oleifera* seeds, *CYCD3-1* showed different trends. *CYCD3-1-1* was increased at first but then decreased, while *CYCD3-1-2* was significantly decreased. This suggested that *CYCD3-1* have different functions in regulating seed size of *C. oleifera*.

The cell expansion was closely related to the stretch in the cell wall. Among them, XTHs and expansins can loosen cell walls and affect cell expansion [44,45]. During the development of *C. oleifera* seeds, most of the *XTHs* and *Expansins* genes were upregulated from the T1 to T3 period, then decreased. Besides, *XTH23* show different treads. The protein sequences of *XTH23* in pepper [46], cucumber [47], *Arabidopsis* [48] and *C. oleifera* of these species were aligned to construct a phylogenetic tree (electronic supplementary material, figure S1). *XTH23-1* in *C. oleifera* was similar to *CaXTH23*. In pepper, *CaXTH23* was highly expressed in the early stages of fruit development and then decreased gradually with fruit ripening [46], which was consistent with the expression trend of *XTH23-1*. *XTH23-3* was a close relative of *CsXTH23*, which was related to the rapid expansion of cells in cucumber fruit spines [47]. In *C. oleifera* seeds, the expression of *XTH23-3* increased 420-fold in the T4 period compared with the T1 period in *C. oleifera* seeds. Combined with the expression levels of *XTH23-3* in *C. oleifera* seeds, it was speculated that it may play an important role in seed development at the nearly mature stage. In *Arabidopsis*, *XTH9* was involved in the expansion of wood cells [49]. *XTH9* was highly expressed in the T1 period of *C. oleifera* seeds and then decreased, which may be related to vigorous cell expansion. Additionally, *Expansins* genes were involved in cell wall relaxation and extension in *Arabidopsis* seeds [50]. Suppression of *EXPA1* and *EXPA10* altered chemical composition of the cell wall [51]. In *C. oleifera* seeds, most of the *Expansins* genes were increased first (T1–T3) but then significantly decreased. It was speculated that *Expansins* genes mainly act on the period of rapid expansion of *C. oleifera* seeds.

In addition to cell proliferation and cell expansion, the ubiquitin-proteasome pathway, G-protein signalling pathway, MAPK pathway and IKU pathway were identified to be involved in the regulation of seed size [26]. In the ubiquitin-proteasome pathway, *DA1* negatively regulates seed size [52]. The larger seeds produced by the *da1-1* mutant was the result of enlargement of sporophyte integument in *Arabidopsis* [53]. In *C. oleifera* seeds, the expression levels of *DA1* were reduced by 86%, which was consistent with the negative regulation of seed size by *DA1* in *Arabidopsis*. Among the genes screened in the MAPK pathway, *MKP1* control grain size by restricting cell proliferation in grain hulls and the overexpression of *MKP1* leads to small grains in rice [54]. In *C. oleifera* seeds, the expression levels of *MKP1* were downregulated by 2.7-fold in the T5 period. Unlike the ubiquitin-proteasome pathway, the G-protein signalling pathway and MAPK pathway regulated seed size by maternal tissues, the IKU pathway controls seed size by regulating zygotic tissues [26,27]. In *Arabidopsis*, *iku1* and *iku2* produced small seeds and led to early cellularization of endosperm, which was similar to the seeds with excessive parental dosage [55]. In the development of *C. oleifera* seeds, *IKU1* and *IKU2* were downregulated. Combined with *Arabidopsis*, *IKU1* and *IKU2* were important participants in the development of endosperm.

## 4.2. Role of phytohormones in regulating Camellia oleifera seed size

Phytohormone regulates diverse processes in seed growth and development. Auxin, BR and SA have been suggested to play an important role in the regulation of seed size control [28]. In *Arabidopsis*, *YUC4* is a key gene in the process of auxin biosynthesis, and the overexpression of *YUC4* influenced

auxin homeostasis and seed size [31,56]. In the process of *C. oleifera* seed development, the expression level of *YUC4* was 1724-fold in T1 compared with T5, and its significant downregulation may be related to the accumulation of auxin. *AUX/IAA* and *ARF* suppress and regulate auxin signal transduction [57]. Among them, *IAA14* was involved in the formation of *Arabidopsis* lateral roots [58] and *IAA17* control tomato fruit size via the regulation of cell size [59]. In *C. oleifera* seeds, *IAA14* was upregulated 799-fold in T4 compared with that in T1. The expression level of *IAA17* was reduced by 98% from the TI to T5 period. It suggested that *IAA14* and *IAA17* regulate *C. oleifera* seed size by participating in auxin signal transduction. *ARF2* inhibits the proliferation of maternal cells [31,60]. In *C. oleifera* seeds, *ARF2* was downregulated and significantly correlated with seed phenotype, which indicates that *ARF2* negatively regulates seed size in *C. oleifera* as in *Arabidopsis*. Additionally, studies on the change of SA-related genes of *C. oleifera* seeds showed that *PAL-2*, *PAL1* and *PAL6* in the SA biosynthesis pathway were significantly downregulated from the T1 to T5 period, and the gene expression of the signalling pathway did not change significantly in *C. oleifera* seeds. The downregulation of *PAL-2*, *PAL1* and *PAL6* in *C. oleifera* seeds may be related to the formation of secondary metabolites [61]. In *Arabidopsis*, *BZR1* is an important TF for regulating organ size in the BR signalling pathway. The overexpression of *ZmBZR1* in *Arabidopsis* shows seed enlargement [62], however, *BZR1* had not significantly changed in *C. oleifera* seeds.

## 4.3. The regulation of transcription factors in *Camellia oleifera* seed size

TFs were also involved in the regulation of seed size [26]. First, the hub genes related to the three traits of *C. oleifera* seed were analysed by WGCNA. Among the screened hub genes, the *SPL* gene family plays an important role in plant growth and increases crop yield by regulating the architecture of rice plants [63,64]. In rice, *SPL4* could be a key regulator of grain size by acting on cell division control to improve rice yield and the overexpression of *SPL4* decreased grain weight and size [65]. Seed-specific overexpression of *SPL12* and *plant Architecture 1* (*IPA1*) enhance seed improved seed dormancy grain size in rice [66]. In *C. oleifera* seeds, *SPL4* and *SPL12* were decreased to 1% and 36% in T5, respectively, which indicates *SPL4* and *SPL12* may be important participants in the regulation of seed size. Additionally, *WRKY44* controls final seed size by regulating seed coat development in *Arabidopsis* [67]. *FAR1* affected cell proliferation by regulating the cell cycle, and its transcription in the G1 phase was essential for protein accumulation and pheromone-induced cell cycle arrest [68]. In *C. oleifera* seeds, *WRKY44* and *FAR1* were decreased to 21% and 24% in the T5 period, respectively. *WRKY44* maybe related to seed coat development and *FAR1* may be correlated with cell proliferation in *C. oleifera* seed development. Among the selected hub genes related to horizontal diameter, *ABI4* and *YAB1* were closely associated with *Camellia* seed size. The overexpression of *ABI4* decreased in plant height and poor seed production in *Arabidopsis* [69]. The rice *YAB1* gene causes dwarfing and affects yield [70]. In *C. oleifera* seeds, *ABI4* and *YAB1* were significantly increased 800-fold in T3 and 560-fold in T4, respectively. It indicated that *ABI4* and *YAB1* may be important participants in the regulation of *C. oleifera* seed size.

The expression levels of known seed size-related TFs were also screened in *C. oleifera* seeds [26]. Among them, *AP2* affected the development of zygotic embryos and endosperm to inhibit seed size of *Arabidopsis*. The *ap2* mutant produced larger seeds than the wild-type [71]. *KLU* regulated seed size by promoting the proliferation of maternal cells [72]. *EOD3* could act maternally to promote cotyledon cell expansion and thus affects the size of rapeseed [73]. In *C. oleifera* seeds, *AP2* and *EOD3* were significantly increased 14-fold and 259-fold in the T5 period, respectively, while *KLU* significantly decreased (2040-fold), which suggests that these genes may play important roles in the regulation of the *C. oleifera* seed size.

# 5. Conclusion

During seed development of *C. oleifera*, the seeds horizontal diameter, vertical diameter and volume showed rapid growth from 235 DAP (T2) to 258 DAP (T3) with a slight increase at seed maturity (T4–T5). The content of IAA, SA and BR showed an increasing trend from the T1 to T3 period, but then decreased at seed maturity (T4–T5). By studying genes related to hormone biosynthesis and signalling pathways, we found that most genes related to the auxin signalling pathway were significantly negatively correlated with seed horizontal diameter, vertical diameter and volume. Among them, *YUC4* was significantly downregulated (1724-fold). Through WGCNA, 21 hub genes

were screened. Among them, *SPL4* was significantly decreased and associated with seed horizontal diameter, vertical diameter and volume. *ABI4* and *YAB1* were significantly increased 800-fold in T3 and 560-fold in T4, respectively. Additionally, *KLU* significantly decreased (2040-fold). Taken together, these results suggest that these genes are important participants in regulating the size of *C. oleifera* seeds.

Data accessibility. The Illumina RNA-Seq data of *Camellia oleifera* seeds were uploaded to SRA with accession no. PRJNA668531 (https://dataview.ncbi.nlm.nih.gov/object/PRJNA668531).

Authors' contributions. K.J.: conceptualization, data curation, formal analysis, investigation, methodology, software, validation, writing—original draft and writing—review and editing; Q.S.: conceptualization, investigation, methodology, resources and supervision; X.Y.: conceptualization, data curation, formal analysis and methodology; C.T.: investigation, project administration and resources; L.W.: data curation, methodology and software; L.C.: data curation and software; X.X.: data curation and software; W.G.: conceptualization, data curation, funding acquisition, methodology, project administration, resources and writing—review and editing; D.Y.: conceptualization, funding acquisition, project administration, resources and writing—review and editing.

All authors gave final approval for publication and agreed to be held accountable for the work performed therein.

Competing interests. We declare we have no competing interests.

Funding. This work was supported by the National Key R&D Program of China (grant no. 2018YFD1000603-1), the Scientific Research Foundation for Advanced Talents of Central South University of Forestry and Technology (grant no. 2018YJ002), the Postgraduate Scientific Research Innovation Project of Hunan Province (Effect of exogenous methyl jasmonate on flavonoid synthesis in *Camellia oleifera* seeds), and the Natural Science Foundation of Hunan Province (2020JJ5968).

Acknowledgement. We thank Hannah Holmberg of Louisiana State University, Baton Rouge, for her help in revising the manuscript.

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
