## [Peer Review File · Royal Society Open Science]

Review History

RSOS-211138.R0 (Original submission)

Review form: Reviewer 1

Is the manuscript scientifically sound in its present form?

No

Are the interpretations and conclusions justified by the results?

Yes

Is the language acceptable?

No

Do you have any ethical concerns with this paper?

No

Have you any concerns about statistical analyses in this paper?

No

Recommendation?

Major revision is needed (please make suggestions in comments)

Comments to the Author(s)

In this study entitled "Hormone analysis and candidate genes identification associated with seed size in *Camellia oleifera*", the authors investigated seed size manipulated by phytohormone signaling pathway genes and transcriptional factors mainly through phenotypic observing, gene expression pattern exploring, and weighted gene correlation network analysis (WGCNA). The fast seed size growth was from 235 DAP to 258 DAP in camellia cultivars. Auxin signal pathway may play a key role in negatively regulating seed size. Through WGCNA, a total of 21 hub genes were screened out. Among them, *SPL4*, *ABI4*, *YAB1*, and *KLU* may be important participants in regulating the seed size.

This research work proposed molecular regulation mechanism of seed size that provides a theoretical basis for improving the yield of *C. oleifera* seeds. However, this manuscript has many details to be improved. Only after substantial revision, this manuscript could be accepted.

Major concern:

- 1, In results 4.2, How the authors acquired these gene expression data? In the Materials and methods section, there is no description of transcriptional expression data acquisition. Please add the processes of gene expression data acquisition.
- 2, In results 4.3, Why cell wall-related genes need to be investigated, please clarify. From 4.1 to 4.2, the authors simply described the phenotypic and gene expression data but did not conclude any brief summaries from these data, which makes readers hardly understand the results.
- 3, In results 4.5, the authors list the data of phytohormone contents and genes expression patterns in the signaling pathway, especially including the relationship between these gene expression patterns and hormone signal pathways. From my perspective, these data can not offer us sufficient information to understand how these genes and hormones regulating *Camellia* seed development. The authors need to introduce more background knowledge of phytohormones signaling pathways.
- 4, In results 4.6, where the TFs and transcriptome data come from? Please add the data and substantial analysis process.
- 5, In the discussion part, 5.1, It seems that the authors believe T1 to T3 stages are mainly controlled by cell proliferation, while T4 to T5 stages are manipulated by cell expansion. Where this recognition comes from? Please clarify it.
- 6, In 5.2, The authors demonstrated that auxin plays a key role in *Camellia* seed size. But the authors did not discuss salicylic acid and brassinolide signaling pathways. Therefore, I suggest the authors discuss all the detected hormones comprehensively.

Minor concern:

- 1, In summary, Line 29, To make the summary can be more fluent and easier to understand, it would be better to change "seed yield" into "yield of seed oil".
- 2, In the Introduction part, Line 46 to 48, "Manipulating seed size is an ideal way to solve this problem, because it is reported that seed size is a crucial factor to influence the final seed size." could be an alternative to "Seed size is..... plants".
- 3, In Line 54, In the description of "their activity is regulated by cyclins and phosphorylation/dephosphorylation", "cyclins" and "phosphorylation/dephosphorylation" are not the same category. Thus modify one of the descriptions or change the "and" into "though" might be good solutions.
- 4, In Line 58, "Plant cell expansion is closely related to the cell wall." This sentence is not clearly linked with the upper paragraph. In the last paragraph, the authors talking about cell proliferation and seed development, but herein the relationship between cell expansion and cell wall was reviewed. I think the authors need to introduce more information between these two paragraphs.

- 5, In the section of Plant materials collection, line 29, the grammar is wrong, and It's better to replace "is" with "come from".
- 6, In section 3.1, Line 60, "camellia" should be regular font.
- 7, In section 3.1, lines 33 to 34, the authors described that "Fifteen trees without pests and diseases, growing well, and consistent management measures were selected randomly". I am really curious that 6-years-old tree without any pests and diseases. If it is true, please explain it. If it is not a reasonable description, please revise it.
- 8, In section 3.2, line 42, "Measure seed volume with drainage method". Please describe the drainage method in detail.
- 9, In section 3.3, lines 48 to 49, please describe the phytohormones content in detail or add a reference.
- 10, In section 3.5, Please add the version of these software and "p" should be italic.
- 11, In results 4.2, Line 26 to 29, To let the readers easily go through your research work, I suggest that authors introduce CYCD3-1-1 and CYCD3-1-2 independently.
- 12, In figure 4, Please improve the DPI of the plot, it is not visible.
- 13, For tables, please keep every independent table on the same page.
- 14, The writing of this manuscript needs to be improved, please find help from some native researchers.

Decision letter (RSOS-211138.R0)

Dear Dr Gong

The Editors assigned to your paper RSOS-211138 "Hormone analysis and candidate genes identification associated with seed size in *Camellia oleifera*" have now received comments from reviewers and would like you to revise the paper in accordance with the reviewer comments and any comments from the Editors. Please note this decision does not guarantee eventual acceptance.

Please submit your revised manuscript and required files (see below) no later than 21 days from today's (ie 22-Oct-2021) date. Note: the ScholarOne system will 'lock' if submission of the revision is attempted 21 or more days after the deadline. If you do not think you will be able to meet this deadline please contact the editorial office immediately.

Please note article processing charges apply to papers accepted for publication in Royal Society Open Science (<https://royalsocietypublishing.org/rsos/charges>). Charges will also apply to papers transferred to the journal from other Royal Society Publishing journals, as well as papers

submitted as part of our collaboration with the Royal Society of Chemistry (<https://royalsocietypublishing.org/rsos/chemistry>). Fee waivers are available but must be requested when you submit your revision (<https://royalsocietypublishing.org/rsos/waivers>).

on behalf of Dr James Locke (Associate Editor) and Malcolm White (Subject Editor)
openscience@royalsociety.org

Associate Editor Comments to Author (Dr James Locke):

Comments to the Author:

The reviewer requires major revisions before acceptance and the improvements requested are reasonable. Please correct all the points raised in the revision.

Reviewer comments to Author:

Reviewer: 1

Comments to the Author(s)

In this study entitled “Hormone analysis and candidate genes identification associated with seed size in *Camellia oleifera*”, the authors investigated seed size manipulated by phytohormone signaling pathway genes and transcriptional factors mainly through phenotypic observing, gene expression pattern exploring, and weighted gene correlation network analysis (WGCNA). The fast seed size growth was from 235 DAP to 258 DAP in camellia cultivars. Auxin signal pathway may play a key role in negatively regulating seed size. Through WGCNA, a total of 21 hub genes were screened out. Among them, SPL4, ABI4, YAB1, and KLU may be important participants in regulating the seed size.

This research work proposed molecular regulation mechanism of seed size that provides a theoretical basis for improving the yield of *C. oleifera* seeds. However, this manuscript has many details to be improved. Only after substantial revision, this manuscript could be accepted.

Major concern:

- 1, In results 4.2, How the authors acquired these gene expression data? In the Materials and methods section, there is no description of transcriptional expression data acquisition. Please add the processes of gene expression data acquisition.
- 2, In results 4.3, Why cell wall-related genes need to be investigated, please clarify. From 4.1 to 4.2, the authors simply described the phenotypic and gene expression data but did not conclude any brief summaries from these data, which makes readers hardly understand the results.
- 3, In results 4.5, the authors list the data of phytohormone contents and genes expression patterns in the signaling pathway, especially including the relationship between these gene expression patterns and hormone signal pathways. From my perspective, these data can not offer us sufficient information to understand how these genes and hormones regulating *Camellia* seed development. The authors need to introduce more background knowledge of phytohormones signaling pathways.
- 4, In results 4.6, where the TFs and transcriptome data come from? Please add the data and substantial analysis process.
- 5, In the discussion part, 5.1, It seems that the authors believe T1 to T3 stages are mainly controlled by cell proliferation, while T4 to T5 stages are manipulated by cell expansion. Where this recognition comes from? Please clarify it.

6, In 5.2, The authors demonstrated that auxin plays a key role in Camellia seed size. But the authors did not discuss salicylic acid and brassinolide signaling pathways. Therefore, I suggest the authors discuss all the detected hormones comprehensively.

Minor concern:

- 1, In summary, Line 29, To make the summary can be more fluent and easier to understand, it would be better to change "seed yield" into "yield of seed oil".
- 2, In the Introduction part, Line 46 to 48, "Manipulating seed size is an ideal way to solve this problem, because it is reported that seed size is a crucial factor to influence the final seed size." could be an alternative to "Seed size is..... plants".
- 3, In Line 54, In the description of "their activity is regulated by cyclins and phosphorylation/dephosphorylation", "cyclins" and "phosphorylation/dephosphorylation" are not the same category. Thus modify one of the descriptions or change the "and" into "though" might be good solutions.
- 4, In Line 58, "Plant cell expansion is closely related to the cell wall." This sentence is not clearly linked with the upper paragraph. In the last paragraph, the authors talking about cell proliferation and seed development, but herein the relationship between cell expansion and cell wall was reviewed. I think the authors need to introduce more information between these two paragraphs.
- 5, In the section of Plant materials collection, line 29, the grammar is wrong, and It's better to replace "is" with "come from".
- 6, In section 3.1, Line 60, "camellia" should be regular font.
- 7, In section 3.1, lines 33 to 34, the authors described that "Fifteen trees without pests and diseases, growing well, and consistent management measures were selected randomly". I am really curious that 6-years-old tree without any pests and diseases. If it is true, please explain it. If it is not a reasonable description, please revise it.
- 8, In section 3.2, line 42, "Measure seed volume with drainage method". Please describe the drainage method in detail.
- 9, In section 3.3, lines 48 to 49, please describe the phytohormones content in detail or add a reference.
- 10, In section 3.5, Please add the version of these software and "p" should be italic.
- 11, In results 4.2, Line 26 to 29, To let the readers easily go through your research work, I suggest that authors introduce CYCD3-1-1 and CYCD3-1-2 independently.
- 12, In figure 4, Please improve the DPI of the plot, it is not visible.
- 13, For tables, please keep every independent table on the same page.
- 14, The writing of this manuscript needs to be improved, please find help from some native researchers.

===PREPARING YOUR MANUSCRIPT===

Please ensure that you include an acknowledgements' section before your reference list/bibliography. This should acknowledge anyone who assisted with your work, but does not

qualify as an author per the guidelines at <https://royalsociety.org/journals/ethics-policies/openness/>.

===PREPARING YOUR REVISION IN SCHOLARONE===

-- Ensure that your data access statement meets the requirements at <https://royalsociety.org/journals/authors/author-guidelines/#data>. You should ensure that you cite the dataset in your reference list. If you have deposited data etc in the Dryad repository, please include both the 'For publication' link and 'For review' link at this stage.

Author's Response to Decision Letter for (RSOS-211138.R0)

See Appendix A.

RSOS-211138.R1 (Revision)

Review form: Reviewer 1

Is the manuscript scientifically sound in its present form?

No

Are the interpretations and conclusions justified by the results?

Yes

Is the language acceptable?

No

Do you have any ethical concerns with this paper?

No

Have you any concerns about statistical analyses in this paper?

No

Recommendation?

Major revision is needed (please make suggestions in comments)

Comments to the Author(s)

In this study entitled “Hormone analysis and candidate genes identification associated with seed size in *Camellia oleifera*”, the authors have analyzed transcriptome data from public databases, screened some hub transcription factors and related-hormones pathways for *C. oleifera* seed horizontal diameter, vertical diameter, and volume variation through WGCNA. However, there are several concerns about the interpretation of the data, and some additional analysis and discussion are needed.

Major Issues:

- 1) The overall idea and logic of the “Introduction” section are confusing, so I suggest authors revise it.
- 2) In “3.1. Plant materials collection” section, authors collected “the fruits of 210 DAP (days after 88 pollination; T1), 235 DAP (T2), 258 DAP (T3), 292 DAP (T4) and 333 DAP (T5)” as plant materials, are there any screening criteria for the selection of these four period?
- 3) In “4.1. Variations in *C. oleifera* seeds size at different developmental stages” section, authors should provide more data to prove the result about “the key period of *C. oleifera* seed expansion is from 235 to 258 DAP”, and why is it accurate to 235 and 258 DAP?
- 4) Line 151, “Unlike XTH9 and XTH16, XTH23 showed different trends, XTH23-2 was decreased from T1 to T5 period while the expression of XTH23-3 increased 420-fold in T4 period compared with T1 period.” It is interesting, but it is not discussed in detail in the “Discussion” section, please discuss it more in detail.
- 5) In “4.4. Variations in gene expression in signal pathways associated with the control of seed size” section, it is the most important part of the manuscript, which is really relevant to the topic of the article, but the author has written it too briefly. Please write it in detail.
- 6) Line 185, “most auxin-related genes were negatively correlated with seed horizontal diameter, vertical diameter, and volume of *C. oleifera* seeds”, how to explain this negative regulatory relationship in controlling phenotypes? It seems inconsistent with the previous content.
- 7) In “5.2. Auxin plays a key role in *Camellia* seed size” section, more than Auxin is discussed in detail, so the title should be revised.
- 8) In “5.3. The regulation of TFs in *C. oleifera* seeds” section, it's a very strange result, why not measure the ABA and GA in the previous content, the whole article is quite confused about hormones.

Minor Issues:

- 1) Authors did not specify the biological repeat of hormones detection. In addition, in line 166, the author mentions IAA, BR, and SA three hormones, how to identify these three hormones that were most associated with the phenotype?
- 2) Line 101, “8000 r min⁻¹” should be, 8000 r min⁻¹, please check the writing specification in the article.
- 3) Line 181, gene names should be italicized.
- 4) The use of acronyms is very informal, such as transcription factors (TF), weighted gene correlation network analysis (WGCNA) et. al. please check carefully.
- 5) Manuscripts need to be edited by native English speakers.

Decision letter (RSOS-211138.R1)

Dear Dr Gong

The Editors assigned to your paper RSOS-211138.R1 "Hormone analysis and candidate genes identification associated with seed size in *Camellia oleifera*" have now received comments from reviewers and would like you to revise the paper in accordance with the reviewer comments and any comments from the Editors. Please note this decision does not guarantee eventual acceptance.

Please submit your revised manuscript and required files (see below) no later than 21 days from today's (ie 23-Dec-2021) date. Note: the ScholarOne system will 'lock' if submission of the revision is attempted 21 or more days after the deadline. If you do not think you will be able to meet this deadline please contact the editorial office immediately.

on behalf of Dr James Locke (Associate Editor) and Malcolm White (Subject Editor)
openscience@royalsociety.org

Associate Editor Comments to Author (Dr James Locke):

Associate Editor: 1

Comments to the Author:

The reviewer has examined the revised manuscript and has requested major revisions to the paper. Please complete all their requests, including checking the language used, in a revision.

Reviewer comments to Author:

Reviewer: 1

Comments to the Author(s)

In this study entitled "Hormone analysis and candidate genes identification associated with seed size in *Camellia oleifera*", the authors have analyzed transcriptome data from public databases, screened some hub transcription factors and related-hormones pathways for *C. oleifera* seed horizontal diameter, vertical diameter, and volume variation through WGCNA. However, there

are several concerns about the interpretation of the data, and some additional analysis and discussion are needed.

Major Issues:

- 1) The overall idea and logic of the “Introduction” section are confusing, so I suggest authors revise it.
- 2) In “3.1. Plant materials collection” section, authors collected “the fruits of 210 DAP (days after 88 pollination; T1), 235 DAP (T2), 258 DAP (T3), 292 DAP (T4) and 333 DAP (T5)” as plant materials, are there any screening criteria for the selection of these four period?
- 3) In “4.1. Variations in *C. oleifera* seeds size at different developmental stages” section, authors should provide more data to prove the result about “the key period of *C. oleifera* seed expansion is from 235 to 258 DAP”, and why is it accurate to 235 and 258 DAP?
- 4) Line 151, “Unlike XTH9 and XTH16, XTH23 showed different trends, XTH23-2 was decreased from T1 to T5 period while the expression of XTH23-3 increased 420-fold in T4 period compared with T1 period.” It is interesting, but it is not discussed in detail in the “Discussion” section, please discuss it more in detail.
- 5) In “4.4. Variations in gene expression in signal pathways associated with the control of seed size” section, it is the most important part of the manuscript, which is really relevant to the topic of the article, but the author has written it too briefly. Please write it in detail.
- 6) Line 185, “most auxin-related genes were negatively correlated with seed horizontal diameter, vertical diameter, and volume of *C. oleifera* seeds”, how to explain this negative regulatory relationship in controlling phenotypes? It seems inconsistent with the previous content.
- 7) In “5.2. Auxin plays a key role in *Camellia* seed size” section, more than Auxin is discussed in detail, so the title should be revised.
- 8) In “5.3. The regulation of TFs in *C. oleifera* seeds” section, it's a very strange result, why not measure the ABA and GA in the previous content, the whole article is quite confused about hormones.

Minor Issues:

- 1) Authors did not specify the biological repeat of hormones detection. In addition, in line 166, the author mentions IAA, BR, and SA three hormones, how to identify these three hormones that were most associated with the phenotype?
- 2) Line 101, “8000 r min⁻¹” should be, 8000 r min⁻¹, please check the writing specification in the article.
- 3) Line 181, gene names should be italicized.
- 4) The use of acronyms is very informal, such as transcription factors (TF), weighted gene correlation network analysis (WGCNA) et, al. please check carefully.
- 5) Manuscripts need to be edited by native English speakers.

===PREPARING YOUR MANUSCRIPT===

If you have been asked to revise the written English in your submission as a condition of publication, you must do so, and you are expected to provide evidence that you have received language editing support. The journal would prefer that you use a professional language editing service and provide a certificate of editing, but a signed letter from a colleague who is a fluent speaker of English is acceptable. Note the journal has arranged a number of discounts for authors using professional language editing services (<https://royalsociety.org/journals/authors/benefits/language-editing/>).

===PREPARING YOUR REVISION IN SCHOLARONE===

<https://royalsociety.org/journals/authors/author-guidelines/#supplementary-material> to include a suitable title and informative caption. An example of appropriate titling and captioning may be found at https://figshare.com/articles/Table_S2_from_Is_there_a_trade-off_between_peak_performance_and_performance_breadth_across_temperatures_for_aerobic_scorpions_in_teleost_fishes_/3843624.

Author's Response to Decision Letter for (RSOS-211138.R1)

See Appendix B.

RSOS-211138.R1 (Revision)

Review form: Reviewer 1

Is the manuscript scientifically sound in its present form?

Yes

Are the interpretations and conclusions justified by the results?

Yes

Is the language acceptable?

Yes

Do you have any ethical concerns with this paper?

No

Have you any concerns about statistical analyses in this paper?

No

Recommendation?

Accept as is

Comments to the Author(s)

The authors have addressed all my concerns.

Decision letter (RSOS-211138.R2)

Dear Dr Gong,

It is a pleasure to accept your manuscript entitled "Hormone analysis and candidate genes identification associated with seed size in *Camellia oleifera*" in its current form for publication in Royal Society Open Science. The comments of the reviewer(s) who reviewed your manuscript are included at the foot of this letter.

on behalf of Dr James Locke (Associate Editor) and Malcolm White (Subject Editor)
openscience@royalsociety.org

Associate Editor Comments to Author (Dr James Locke):

The reviewer is happy with the revision and the manuscript is now suitable for publication.

Reviewer comments to Author:

Reviewer: 1

Comments to the Author(s)

The authors have addressed all my concerns.

Appendix A

Dear Editor and Reviewers,

Thank you for your letter and for the reviewers' comments concerning our manuscript entitled "Hormone analysis and candidate genes identification associated with seed size in *Camellia oleifera*" (Manuscript No. RSOS-211138).

Those comments are all valuable and very helpful for revising and improving our paper, as well as the important guiding significance to our researches. We have studied comments carefully and have made correction which we hope meet with approval. Revised portion are marked in **yellow highlight** in the paper. The main corrections in the paper and the responds to the reviewer's comments are as flowing:

Reviewer: 1

Major concern:

1. In results 4.2, How the authors acquired these gene expression data? In the Materials and methods section, there is no description of transcriptional expression data acquisition. Please add the processes of gene expression data acquisition.

Response: Thanks for your suggestion. Our gene expression data were obtained from published transcriptome data in NCBI database. The Illumina RNA-Seq data of *Camellia oleifera* seeds were uploaded to SRA with accession PRJNA668531 (<https://dataview.ncbi.nlm.nih.gov/object/PRJNA668531>).

According to your suggestions, we made corresponding supplements in Section 3.4 (Line 106-108 in revised manuscript) and uploaded the screened gene expression data (Supplementary file 1).

2. In results 4.3, Why cell wall-related genes need to be investigated, please clarify. From 4.1 to 4.2, the authors simply described the phenotypic and gene expression data but did not conclude any brief summaries from these data, which makes readers hardly understand the results.

Response: Thanks for your good suggestion. We have added "The extension of cell wall affects seed size by regulating cell size (Gillaspy et al.,1993; Robert and Marcelo, 2014)." in sections 4.3 (Line 144 in revised manuscript).

According to the reviewers' suggestion, we have supplemented the brief summaries in sections 4.1 to 4.2 (Line 127-128, 141 in revised manuscript).

Gillaspy G, Ben-David H, Gruissem W. 1993. Fruits: A Developmental Perspective. *The Plant cell* 5, 1439–1451. (doi: 10.1105/tpc.5.10.1439)

Robert S, Marcelo C. 2014. Interplay between cell growth and cell cycle in plants, *J. Exp. Bot.* 65(10), 2703–2714. (doi: 10.1093/jxb/ert354)

3. In results 4.5, the authors list the data of phytohormone contents and genes expression patterns in the signaling pathway, especially including the relationship between these gene expression patterns and hormone signal pathways. From my perspective, these data cannot offer us sufficient information to understand how these genes and hormones regulating *Camellia* seed development. The authors need to introduce more background knowledge of phytohormones signaling pathways.

Response: Thanks for your good suggestion, we supplemented the background of BR, IAA and SA signaling pathways in the revised manuscript (Line 60-67 in revised manuscript).

4. In results 4.6, where the TFs and transcriptome data come from? Please add the data and substantial analysis process.

Response: TFs data comes from the study of Gong et al. (2020) (Table S13). According to your suggestion, we added corresponding reference in section 4.6 (Line 195 in revised manuscript). Transcriptome data comes from NCBI database. The Illumina RNA-Seq data of *Camellia oleifera* seeds were uploaded to SRA with accession PRJNA668531 ([https:// dataview.ncbi.nlm.nih.gov/object/PRJNA668531](https://dataview.ncbi.nlm.nih.gov/object/PRJNA668531)).

Gong W, Song Q, Ji K, Gong S, Wang L, Chen L, Zhang J, Yuan D. 2020. Full-Length Transcriptome from *Camellia oleifera* Seed Provides Insight into the Transcript Variants Involved in Oil Biosynthesis. *J. Agric. Food Chem.* 68, 14670–14683. (doi: 10.1021/acs.jafc.0c05381)

5. In the discussion part, 5.1, It seems that the authors believe T1 to T3 stages are

mainly controlled by cell proliferation, while T4 to T5 stages are manipulated by cell expansion. Where this recognition comes from? Please clarify it.

Response: In the section 5.1, we studied the genes related to cell proliferation and cell expansion at different developmental stages of *Camellia oleifera* seeds. The results showed that the expression of most *CDKs* and cyclin genes related to cell proliferation from T1 to T3 was significantly higher than that from T4 to T5 period (Line 223-224 in revised manuscript), and the expression of most *XTHs* and *Expansins* genes related to cell expansion was also up-regulated from T1 to T3 period, then decreased (Line 234-235 in revised manuscript). Based on the above results, the expression of most cell proliferation and cell expansion related genes from T1 to T3 period was significantly higher than that from T4 to T5 period. Therefore, most cell proliferation and cell expansion related genes have a greater impact on T1-T3 period than T4-T5 period.

According to your suggestion, in order to avoid confusion, we have supplemented and modified section 5.1, which is reflected in the addition of discussion on *XTH23*. (Line 234-241 in revised manuscript).

6. In 5.2, The authors demonstrated that auxin plays a key role in *Camellia* seed size. But the authors did not discuss salicylic acid and brassinolide signaling pathways. Therefore, I suggest the authors discuss all the detected hormones comprehensively.

Response: According to your good suggestion, we supplement the discussion on the signal pathways of salicylic acid and brassinolide in section 5.2 (Line 260-266 in revised manuscript).

Minor concern:

1. In summary, Line 29, To make the summary can be more fluent and easier to understand, it would be better to change “seed yield” into “yield of seed oil”.

Response: We agree the reviewer's good advice. According to the reviewer's instruction, we decided to replace “seed yield” by “yield of seed oil”. (Line 20 in revised manuscript)

2. In the Introduction part, Line 46 to 48, “Manipulating seed size is an ideal way to solve this problem, because it is reported that seed size is a crucial factor to influence the final seed size.” could be an alternative to “Seed size is..... plants”.

Response: Thanks for your good suggestion. We decided to replace “Seed size is a crucial factor to influence final seed yield in plants” by “Manipulating seed size is an ideal way to solve this problem, because it is reported that seed size is a crucial factor to influence the final seed size”. (Line 36-37 in revised manuscript)

3. In Line 54, In the description of “their activity is regulated by cyclins and phosphorylation/dephosphorylation”, “cyclins” and “phosphorylation/dephosphorylation” are not the same category. Thus modify one of the descriptions or change the “and” into “though” might be good solutions.

Response: Thanks for your good suggestion. We decided to replace “and” by “through”. (Line 44 in revised manuscript)

4. In Line 58, “Plant cell expansion is closely related to the cell wall.” This sentence is not clearly linked with the upper paragraph. In the last paragraph, the authors talking about cell proliferation and seed development, but herein the relationship between cell expansion and cell wall was reviewed. I think the authors need to introduce more information between these two paragraphs.

Response: Thanks for your good suggestion. We combined the two paragraphs and supplement the relationship between cell wall and cell expansion in revised manuscript. (Line 46-48 in revised manuscript)

5. In the section of Plant materials collection, line 29, the grammar is wrong, and It’s better to replace “is” with “come from”.

Response: Thanks for your good suggestion. We have replaced “is” with “come from”. (Line 81 in revised manuscript)

6. In section 3.1, Line 60, “camellia” should be regular font.

Response: Thanks for your good suggestion. We have revised the format of “camellia” in section 3.1, Line 60. (Line 82 in revised manuscript)

7. In section 3.1, lines 33 to 34, the authors described that “Fifteen trees without pests and diseases, growing well, and consistent management measures were selected randomly”. I am really curious that 6-years-old tree without any pests and diseases. If it is true, please explain it. If it is not a reasonable description, please revise it.

Response: Based on the reviewer’s suggestion, we decided to replace “Fifteen trees without pests and diseases” by “Fifteen trees without significant pests or diseases”. Thanks for your suggestion. (Line 85 in revised manuscript)

8. In section 3.2, line 42, “Measure seed volume with drainage method”. Please describe the drainage method in detail.

Response: Thanks for your good suggestion. Seed volume was measured according to the method of Li et al. Briefly, 50ml of water was added into the 100ml measuring cylinder, and the recording volume was V1. After adding 5 seeds, the recording volume was V2, so the volume of each seed was $V2 - V1 / 5$. When the seeds were small (235 DAP and 258DAP), the seed volume was measured with a 10ml measuring cylinder. In addition, we have made corresponding supplements in part 3.2 of the manuscript. (Line 93-96 in revised manuscript)

Li YG, Liu XH, Ma JW, Shi CG, Zhu GQ. 2014. Phenotypic variations in populations of *Phoebe chekiangensis*. *Chinese J. Plant Ecol.* 38(12), 1315-1324. (doi: 10.3724/SP.J.1258.2014.00126)

9. In section 3.3, lines 48 to 49, please describe the phytohormones content in detail or add a reference.

Response: Thanks for your good instruction. We have added a reference in section 3.3. (Line 99 in revised manuscript)

10. In section 3.5, Please add the version of these software and “p” should be italic.

Response: According to the reviewer's instruction, we added the corresponding version of the software in section 3.5 and modified the format of “p”. (Line 118 in revised manuscript)

11. In results 4.2, Line 26 to 29, To let the readers easily go through your research work, I suggest that authors introduce *CYCD3-1-1* and *CYCD3-1-2* independently.

Response: According to your good suggestion, we modified “*CYCD3-1-1* was increased first but then decreased, while *CYCD3-1-2* was significant decreased. The expression trend of *CYCD3-1-1* was consistent with the size change of *C. oleifera* seeds.” as “The expression of *CYCD3-1-1* increased from T1 to T3, which was consistent with the change trend of seed size however, the expression of *CYCD3-1-2* decreased from T1 to T3, which was opposite to the change trend of seed size.” (Line 138-140 in revised manuscript)

12. In figure 4, Please improve the DPI of the plot, it is not visible.

Response: Thanks for your suggestion, we have improved the DPI of figure 4.

13. For tables, please keep every independent table on the same page.

Response: Thanks for your suggestion. After consideration, we integrated all the tables and uploaded them as supplementary files. The original Table1,2,3 was integrated into Table S5, and the other supplementary tables were explained in the revised manuscript.

14. The writing of this manuscript needs to be improved, please find help from some native researchers.

Response: Thanks for your good suggestion, we sought the help of some native English speakers to revise the manuscript and mentioned it in Acknowledgement.

Appendix B

Dear Editor and Reviewers,

Thanks very much for taking your time to review on our manuscript entitled “Hormone analysis and candidate genes identification associated with seed size in *Camellia oleifera*” (Manuscript No. RSOS-211138). Those comments and suggestions are all valuable and very helpful for revising and improving our manuscript. We have studied comments carefully and tried our best to made corrections. Below list the modifications we have made and the answers to these questions. Revisions in the manuscript are shown using **red highlight**.

Once again, we acknowledge your comments very much, which are valuable in improving the quality of our manuscript.

Sincerely yours,

Wenfang Gong, Ph.D (On behalf of all corresponding authors)

Key Laboratory of Cultivation and Protection for Non-Wood Forest Trees of Ministry of Education and the Key Laboratory of Non-Wood Forest Products of Forestry Ministry, Central South University of Forestry and Technology, Changsha, Hunan, 410004, China.

E-mail: gwf018@126.com

Associate Editor Comments:

The reviewer has examined the revised manuscript and has requested major revisions to the paper. Please complete all their requests, including checking the language used, in a revision.

Reviewer comments:

Reviewer: 1

In this study entitled “Hormone analysis and candidate genes identification associated with seed size in *Camellia oleifera*”, the authors have analyzed transcriptome data from public databases, screened some hub transcription factors and related-hormones pathways for *C. oleifera* seed horizontal diameter, vertical diameter, and volume variation through WGCNA. However, there are several concerns about the interpretation of the data, and some additional analysis and discussion are needed.

Major Issues:

1) The overall idea and logic of the “Introduction” section are confusing, so I suggest authors revise it.

Response: Thanks for your suggestion. The introduction section of the manuscript mainly described some factors that affect the seed size. Firstly, we described the key genes in the processes of cell proliferation and cell expansion during seed development. Then, we described the signal pathways, phytohormones and transcription regulators that have been studied to regulate seed size. Based on your suggestion, we have strengthened the connection between sentences and paragraphs and modified some inappropriate sentences (Line 37-38, 40-47, 52, 54-58, 64-65 in revised manuscript).

2) In “3.1. Plant materials collection” section, authors collected “the fruits of 210 DAP (days after pollination; T1), 235 DAP (T2), 258 DAP (T3), 292 DAP (T4) and 333 DAP (T5)” as plant materials, are there any screening criteria for the selection of these four periods?

Response: In fact, we collected the seeds from 210 DAP to 333DAP. After detecting the seed volume, and the contents of IAA, SA and BR in developmental seeds (as illustrated in following Figure 1-4), we selected the samplings at these five stages (210 DAP, 235 DAP, 258 DAP, 292 DAP and 333 DAP). Thanks for your suggestion.

Figure 1

Figure 2

Figure 3

Figure 4

3) In “4.1. Variations in *C. oleifera* seeds size at different developmental stages” section, authors should provide more data to prove the result about “the key period of *C. oleifera* seed expansion is

from 235 to 258 DAP”, and why is it accurate to 235 and 258 DAP?

Response: In fact, we calculated the horizontal diameter, vertical diameter and seed volume growth rate at different stages of seed development (as illustrated in following Table 1). It was found that the growth rate of seeds from 235 to 258 DAP was significantly higher than that of other stages, so we think this period is the key period of seed growth. Based on your suggestion, we changed “the key period of *C. oleifera* seed expansion is from 235 to 258 DAP” to “235 to 258 DAP was the period of rapid growth of *C. oleifera* seeds” and Table 1 was added to the manuscript as Table S6 (Line 122 in revised manuscript). In addition, our data on the seed volume of *C. oleifera* at development stages also show that 235 to 258 DAP was the period of rapid growth of *C. oleifera* seeds (as illustrated in following Figure 1). Thanks for your suggestion.

Table 1

DAP	horizontal diameter	vertical diameter	seed volume
210-235	18.08%	17.93%	1.22%
235-258	26.67%	23.66%	5.55%
258-292	4.49%	16.75%	4.24%
292-333	1.02%	1.45%	0.76%

Figure 1

4) Line 151, “Unlike *XTH9* and *XTH16*, *XTH23* showed different trends, *XTH23-2* was decreased from T1 to T5 period while the expression of *XTH23-3* increased 420-fold in T4 period compared with T1 period.” It is interesting, but it is not discussed in detail in the “Discussion” section, please discuss it more in detail.

Response: Thanks for your suggestion. We obtained the protein sequences of *XTH23* in cucumber (*Cucumis sativus* L.), pepper (*Capsicum annuum* L.) and *Arabidopsis* (*Arabidopsis thaliana* L.)

from the published papers (Guo et al., 2020; Li et al., 2018; Xu et al., 2020). The protein sequences of *XTH23* in cucumber, pepper, *Arabidopsis* and *Camellia oleifera* of these species were aligned to construct a phylogenetic tree. As shown in the following Figure 5, *XTH23-1* in *Camellia oleifera* was similar to *CaXTH23*. The expression trend of *CaXTH23* in pepper was consistent with that of *XTH23-1* (Line 235-236 in revised manuscript). *XTH23-3* was a close relative of *CsXTH23*, which was related to the rapid expansion of cells in cucumber fruit spines. Combined with the expression levels of *XTH23-3* in *Camellia oleifera* seeds, it was speculated that it may play an important role in the seed development at the nearly mature stage. Based on your suggestion, we added discussion in section 5.1 (Line 233-234, 236-240 in revised manuscript).

Figure 5

Guo P, Chang H, Li Q, Wang L, Ren Z, Ren H, Chen C. 2020. Transcriptome profiling reveals genes involved in spine development during CsTTG1-regulated pathway in cucumber (*Cucumis sativus* L.). *Plant Sci.* (doi: 10.1016/j.plantsci.2019.110354)

Li N, Shen Y, Xia BB, Liang GS, Wu ZM, Liu HQ. 2018. Identification and Expression Analysis of the XTH Gene Family in Pepper (*Capsicum annuum* L.). *Chinese J. Tropical Crops.* 39: 317-324. (doi: 10.3969/j.issn.1000-2561.2018.02.016)

Xu P, Fang S, Chen H, Cai W. 2020. The brassinosteroid-responsive xyloglucan endotransglucosylase/hydrolase 19 (*XTH19*) and *XTH23* genes are involved in lateral root development under salt stress in *Arabidopsis*. *Plant J.* 104(1):59-75. (doi: 10.1111/tpj.14905)

5) In “4.4. Variations in gene expression in signal pathways associated with the control of seed size” section, it is the most important part of the manuscript, which is really relevant to the topic of the article, but the author has written it too briefly. Please write it in detail.

Response: Thanks for your suggestion. We have rewritten section 4.4 (Line 153-162 in revised

manuscript) and added the related discussion in section 5.1 (Line 247-259 in revised manuscript).

6) Line 185, “most auxin-related genes were negatively correlated with seed horizontal diameter, vertical diameter, and volume of *C. oleifera* seeds”, how to explain this negative regulatory relationship in controlling phenotypes? It seems inconsistent with the previous content.

Response: Thanks for your suggestion. As shown in Figure 4Ad, the expression profile of most auxin related genes was negatively correlated with the change trend of horizontal diameter, vertical diameter and seed volume. To explain the results of negative correlation, we listed the expression profiles of auxin-related genes that significantly negatively related to phenotypic data of seeds (as illustrated in following Table 2 and Figure 6). As shown in Table 2, these genes shown three different trends. Eight genes decreased from T1 to T5 (yellow highlight in Table 2); Seven genes decreased in T2, T4, T5 period (orange highlight in Table 2); Three genes were slightly increased and then decreased (green highlight in Table 2). While the horizontal diameter, vertical diameter and volume of seeds showed an upward trend from T1 to T3, and tended to be stable after T4 and T5 (as illustrated in following Table 2). Therefore, in terms of data, the change trend of these genes was generally opposite to that of seed phenotype. However, the expression levels of these genes from T1 to T3 were significantly higher than those in T4-T5. Based on the data of gene expressions and phenotypic changes, it is negative correlation but not a negative regulatory relationship between most auxin-related genes and seed phenotypes.

Table 2

	T1	T2	T3	T4	T5
horizontal diameter	3.79	8.31	14.44	15.97	16.38
vertical diameter	5.19	9.67	15.12	20.81	21.4
Seed volume	0.04	0.34	1.62	3.06	3.37
IAA content	11.85	15.15	21.5	6.38	7.04
IAA9-2	40.77	20.99	19.74	10.37	4.37
IAA13	28.23	6.13	4.53	1.28	1.37
ARF19	10.87	6.96	5.32	4.88	3.30
TIR1	13.05	8.71	7.27	7.60	1.36
YUC4	22.99	0.73	0.70	0.66	0.01
IAA27-1	115.82	68.27	49.43	17.70	4.25
ARF4-1	17.62	9.23	9.04	5.46	3.59
ARF5	26.64	10.19	4.40	1.86	0.94
IAA17	415.42	131.21	166.71	44.08	8.67
ARF1-1	33.97	20.04	24.37	18.12	17.36
ARF2	35.03	12.38	12.46	7.14	2.74
ARF9-1	17.86	11.40	12.94	2.36	0.61
ARF11-2	17.59	12.34	13.71	2.41	0.58

TAR2	24.20	12.34	15.98	3.03	2.11
IAA9-1	110.60	74.07	92.50	51.55	39.55
IAA27-2	117.92	123.88	132.23	46.33	16.18
ARF1-2	16.68	19.40	20.93	12.95	8.12
ARF8	10.68	11.70	10.55	6.66	6.14

Figure 6

7) In “5.2. Auxin plays a key role in *Camellia* seed size” section, more than Auxin is discussed in detail, so the title should be revised.

Response: Thanks for your suggestion. We have changed “Auxin plays a key role in *Camellia* seed size” as “Role of phytohormones in regulating *C. oleifera* seed size” (Line 261 in revised manuscript).

8) In “5.3. The regulation of TFs in *C. oleifera* seeds” section, it's a very strange result, why not measure the ABA and GA in the previous content, the whole article is quite confused about hormones.

Response: Actually, the content of GA and ABA have been published in previous paper (Song et al., 2021). In *Camellia oleifera* seeds, ABA and GA contents were high in the expansion period of

young fruit and relevant with seed development (Song et al., 2021). However, we cannot show these data in our study and only added related contents in the discussion part. Based on your suggestion and in order to avoid confusion, we have modified the part 5.3 in revised manuscript (Line 294-298). Thanks for your suggestion.

Song QL, Ji K, Mo WJ, Wang LK, Chen L, Gao L, Gong WF, Yuan DY. 2021. Dynamics of sugars, endogenous hormones, and oil content during the development of *Camellia oleifera* Abel. fruit. *Botany*. (doi: 10.1139/cjb-2021-0019)

Minor Issues:

1) Authors did not specify the biological repeat of hormones detection. In addition, in line 166, the author mentions IAA, BR, and SA three hormones, how to identify these three hormones that were most associated with the phenotype?

Response: Thanks for your suggestions, we added “Repeat three times” in part 3.3 (Line 97 in revised manuscript) and “All data had three biological replicates” in part 3.5 (Line 112 in revised manuscript). In fact, we determined the contents of several hormones in *Camellia oleifera* seeds. Excluding the published contents of ABA and GA (Song et al., 2021), the contents of IAA, BR and SA were closely related to the change trends of seed phenotype. In addition, IAA and BR has been studied to regulate seed size in *Arabidopsis* and Rice (Li et al., 2019; Li and Li, 2015). SA increased rice yield and regulated cell growth in *Arabidopsis* (Tavares et al., 2014; Miura et al., 2010). Therefore, we chose these three hormones for research.

Song QL, Ji K, Mo WJ, Wang LK, Chen L, Gao L, Gong WF, Yuan DY. 2021. Dynamics of sugars, endogenous hormones, and oil content during the development of *Camellia oleifera* Abel. fruit. *Botany*. (doi: 10.1139/cjb-2021-0019)

Li N, Xu R, Li Y. 2019. Molecular Networks of Seed Size Control in Plants. *Annu. Rev. Plant Biol.* 70, 435–463. (doi: 10.1146/annurev-arplant-050718-095851)

Li N, Li Y. 2015. Maternal control of seed size in plants. *J. Exp. Bot.* 66, 1087–1097. (doi: 10.1093/jxb/eru549)

Tavares LC, Rufino CA, Oliveira Sd, Brunes AP, Villela FA. 2014. Treatment of rice seeds with salicylic acid: seed physiological quality and yield. *J. Seed Sci.* 36, 352–356. (doi:10.1590/2317-1545v36n3636)

Miura K, Lee J, Miura T, Hasegawa PM. SIZ1 controls cell growth and plant development in *Arabidopsis* through

2) Line 101, “8000 r min⁻¹” should be, 8000 r min⁻¹, please check the writing specification in the article.

Response: Thanks for your suggestion, we changed “8000 r min⁻¹” to “8000 r min⁻¹” and checked the manuscript carefully (Line 96 in revised manuscript).

3) Line 181, gene names should be italicized.

Response: Thanks for your suggestion, we checked the format of all gene names in the manuscript and revised them accordingly.

4) The use of acronyms is very informal, such as transcription factors (TF), weighted gene correlation network analysis (WGCNA) et, al. please check carefully.

Response: Thanks for your suggestion, we checked the manuscript carefully and revised them accordingly. “Weighted gene correlation network analysis (WGCNA)” was replaced by “Weighted gene co-expression network analysis (WGCNA)” (Line 24,70 in revised manuscript). “transcription factors (TF)” was replaced by “transcription factors (TFs)” (Line 25 in revised manuscript).

5) Manuscripts need to be edited by native English speakers.

Response: Thanks. We have tried to our best to improve the language in the revised manuscript. In addition, we also invite native English speaker, Dr Arbi J. Sarkissian of Bangor University and Hannah Holmberg of Louisiana State University, Baton Rouge to polish the language thoroughly.